# A Convolutional Neural Network Architecture for Auto-Detection of Landslide Photographs to Assess Citizen Science and Volunteered Geographic Information Data Quality

**Recep Can [1], Sultan Kocaman [1] and Candan Gokceoglu [2],\***

1   Hacettepe University, Department of Geomatics Engineering, 06800 Beytepe, Ankara, Turkey
2   Hacettepe University, Department of Geological Engineering, 06800 Beytepe, Ankara, Turkey
\*   Correspondence: cgokce@hacettepe.edu.tr; Tel.: +90-532-4730851

**Abstract:** Several scientific processes benefit from Citizen Science (CitSci) and VGI (Volunteered Geographical Information) with the help of mobile and geospatial technologies. Studies on landslides can also take advantage of these approaches to a great extent. However, the quality of the collected data by both approaches is often questionable, and automated procedures to check the quality are needed for this purpose. In the present study, a convolutional neural network (CNN) architecture is proposed to validate landslide photos collected by citizens or nonexperts and integrated into a mobile- and web-based GIS environment designed specifically for a landslide CitSci project. The VGG16 has been used as the base model since it allows finetuning, and high performance could be achieved by selecting the best hyper-parameters. Although the training dataset was small, the proposed CNN architecture was found to be effective as it could identify the landslide photos with 94% precision. The accuracy of the results is sufficient for purpose and could even be improved further using a larger amount of training data, which is expected to be obtained with the help of volunteers.

**Keywords:** landslide; convolutional neural network; CitSci; VGI; data quality

## 1. Introduction

With the recent developments in mobile and geospatial technologies, it has become possible to incorporate human efforts in several scientific processes. This approach has brought the term Citizen Science (CitSci) into the sight of researchers and is especially useful for geoscience studies, where every human can act as a powerful sensor and interpreter. The term VGI (volunteered geographical information) has also been introduced, where volunteer contributions to geodata collection have brought about revolutionary changes in this field, although the contributions are often not scientific-achievement-oriented.

The paradigms in landslide research have shown drastic changes depending on the advancements in artificial intelligent algorithms during the last decade e.g., [1–3]. Additionally, CitSci has also become one of the most prominent scientific approaches. The trends in open data and open science mentalities strongly support these processes. In fact, these changes are not only fostering rapid scientific development, but are actually forming science-oriented societies. The Oxford English Dictionary described citizen science as: "Scientific work undertaken by members of the general public, often in collaboration with or under the direction of professional scientists and scientific institutions." [4]. The growing public demand in CitSci [5] can be confirmed by the establishment of several regional and international citizen science associations (e.g., CSA [4], ECSA [6], etc.) in recent years. On the other hand, geodata collection and map production efforts have been increasingly carried

out by nonprofessionals [7]. The term VGI, proposed by Goodchild [8], is commonly used to describe this situation and mainly emphasizes the user-generated geographic information. According to Haklay [9], CitSci stands out as a class of VGI activities that require special attention and analysis. It has been seen in the literature that the terms "CitSci" and "VGI" are used alternatively for many geographical data-related projects, and they also benefit from each other [10]. The development of data quality assessment and validation strategies both for VGI and CitSci data has been an emerging research topic and is crucial, as emphasized by several studies [11–14].

Kocaman et al. [15] have categorized the potential contributions of CitSci to disaster management by analyzing the National Climate Assessment Report [16]. The categories have later been modified for pre-disaster, during, and also post-disaster stages, and the levels of participation in landslide-CitSci (LS-CitSci) projects have been introduced [17]. The levels are (a) basic data collection; (b) data validation and interpretation; (c) advanced interpretation; and (d) project and policy development [17]. Kocaman and Gokceoglu [18] have emphasized the importance and contribution of CitSci for landslide researchers and developed an app, namely LaMA (Landslide Monitoring Application), for landslide data collection. Similarly, the landslide report website [19] by NASA started in the first quarter of 2018 and is a tool for volunteers to provide data about landslides.

LaMA, developed by Kocaman and Gokceoglu [18], is a mobile app that works on iOS and Android platforms and aims at collecting landslide data from volunteers via a simple interface that includes photos of the landslide event, time of occurrence, damage description, and other observations. Data uploaded by citizens using LaMA are stored in a spatial database together with the geolocation information on a central server and are visualized dynamically on a web GIS interface (www.geocitsci.com). Although volunteers can currently upload landslide information, the main question to be answered is the veracity of the collected data, since no quality check is being made on the user side. However, successful data validation would be beneficial for the second stage achievements of LS-CitSci and would also support the data interpretation. In addition, assessment of the landslide photos taken by ordinary people could be extremely difficult due to a lack of expertise and a manual check on the server by the experts prevents auto-run of the system. Depending on the increase in participation of citizens in LS-CitSci projects, the amount of data will be gigantic, and auto-check algorithms for such data are indispensable. Considering this necessity, it is possible to state that the auto-check algorithms for controlling the data provided by citizens are mandatory before they are processed. For this reason, the purpose of the study is to develop a convolutional neural network (CNN) algorithm to validate the landslide photos provided by citizens or nonexperts. For the purpose of the study, a CNN architecture is proposed and executed on the server. The proposed algorithm is trained using well-known image databases and the photos collected from search engines and social media platforms (Figure 1).

Although advanced machine learning algorithms have been widely used for landslide susceptibility mapping studies e.g., [1–3], the use of neural networks for the identification of landslide photos and as support for the data quality check of data collected by citizens or nonprofessionals is new and forms the novelty of the study. The same difficulties, described by Gu et al. [20], are encountered when using the CNN algorithm in this study, which requires big data during the training stage. The main difficulty is to provide a sufficient number of landslide images and further attempts have been carried out to avoid the overfitting problem.

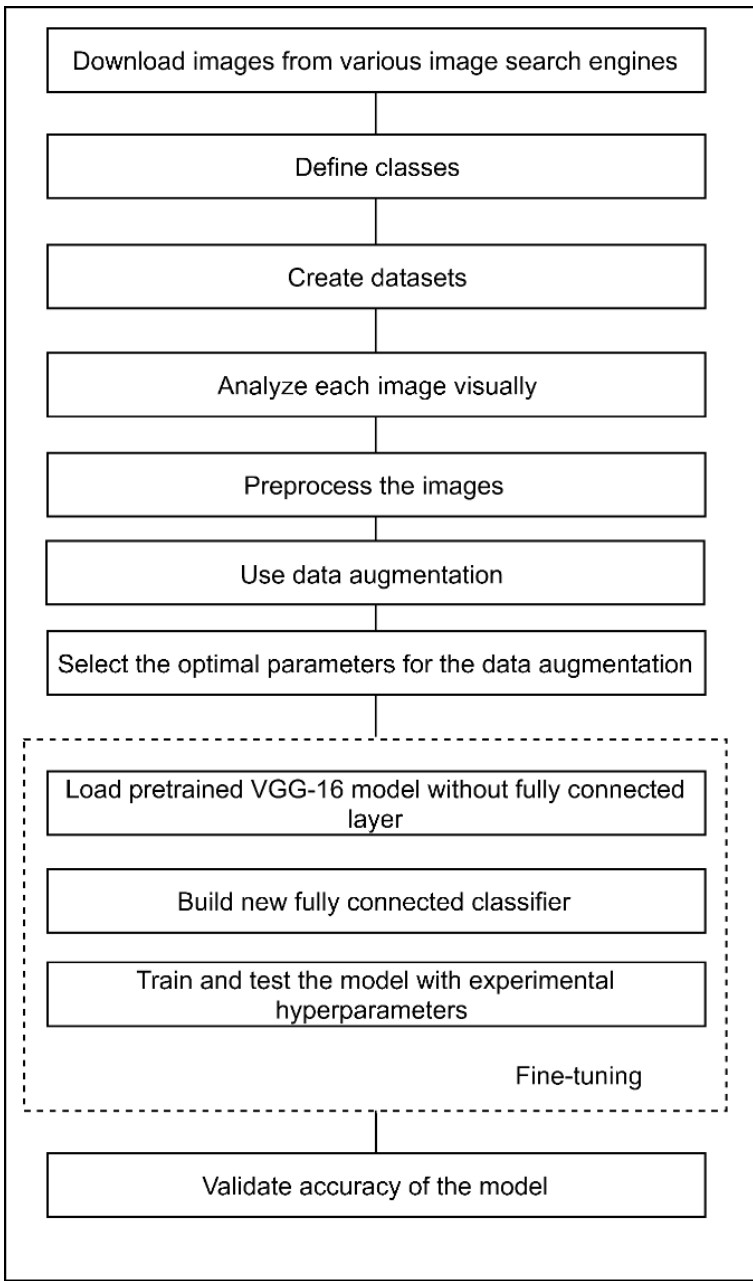

**Figure 1.** General workflow of the study.

## 2. Previous Studies on the Use of CNN in Geosciences

The popularity of CNN has increased recently and this methodology has been successfully applied to various areas [20]. Although many deep learning (DL) algorithms have been developed due to the extreme diversity of its application areas, CNN is perhaps the most widely applied method to extract reasonable information from huge datasets. Object detection, pose estimation, text recognition and detection, speech processing, and natural language processing are some application areas of CNN. However, the major drawbacks of CNN algorithms are: (i) Requirement of large-scale datasets and high computational resources for training; (ii) collecting and labelling the datasets is possible with extensive human efforts; (iii) further research on optimization methods is needed, since selecting the suitable hyper-parameters (e.g., learning rate, kernel sizes, the number of layers) is only possible with skill and experience; and (iv) solid theory of CNN is not yet established, although many successful applications based on it have been found [20].

It is possible to find a massive number of CNN studies in the international literature. Fu and Aldrich [21] applied CNN to flotation froth image analysis. Dung et al. [22] presented a robust method for crack detection by classification in joints of steel bridges and discovered that transfer learning methods improved both the accuracy and robustness of VGG16. Zhang et al. [23] applied a near-to-far learning strategy in a CNN model and improved the accuracy of terrain segmentation and increased robustness against wild environments. The study by Zhang et al. [23] concluded that traditional machine learning techniques have a low ability to generalize and to learn new environments.

Smart and effective energy management systems require reliable load forecasting. For this reason, Sadaei et al. [24] suggested combining fuzzy time series and CNN for short-term load forecasting and they obtained successful results. Bayr and Puschmann [25] applied CNN classification to landscape photographs for detecting woody regrowth vegetation and emphasized the contribution of a robust CNN algorithm in landscape monitoring together with satellite imagery and field measurements. Deng et al. [26] applied a faster region-based CNN to classify objects with multiple classes in satellite images. Coulibaly et al. [27] applied CNN to identify mildew disease in pearl and developed an approach using transfer learning and feature extraction. Mitra et al. [28] trained a CNN-based system to identify six species from microscope digital images and concluded that CNN architecture can provide the 'brain' for a viable robotic picking system. Imamverdiyev and Sukhostat [29] published an interesting geological application of CNN for lithofacies classification by employing different variables. Palafox et al. [30] automatically detected geological landforms on Mars and they applied CNN successfully. Digital images of Berea sandstone samples were used by Karimpouli and Tahmasebi [31] to evaluate the permeability, P- and S-wave velocities, and the formation factor. They found that CNNs demonstrate outstanding performance in predicting the physical parameters of rock if sufficient input data are available. Wu et al. [32] predicted the permeability from images when applying CNN and presented a procedure using CNN with simulated data.

One of the important topics of landslide studies is susceptibility mapping. Despite the fact that until now, several methodologies and approaches have been applied to this field, the use of CNN is rare. Wang et al. [33] produced a landslide susceptibility map of Yanshan County, China, using CNN. Following heavy precipitations and floods, the safety of transportation networks is of vital importance. For this reason, Ahmad et al. [34] proposed an approach to determine passable roads after floods in satellite images and the data from different social media platforms. According to Ahmad et al. [33], the results demonstrated that CNN performs significantly better in comparison to other state-of-the-art approaches. Liu and Wu [35] applied CNN for landslide detection on optical satellite images by performing a 2D wavelet transformation for preprocessing in order to obtain more distinct features and to reduce the image sizes.

In an extensive review on DL applications in geochemical mapping by Zuo et al. [36], it has been emphasized that this field requires more research that implements machine learning and/or DL. Another recent review on the use of machine learning techniques for river flow forecasting published by Yaseen et al. [37] underlined the need for developing more reliable and intelligent expert systems in real-time river flow predictions. According to Grekousis [38], artificial neural networks (ANNs) have a great potential for research in urban geography. Grekousis also proposed a guideline for ANN reports to ensure consistency and to ease the dissemination of the outcomes.

Huang et al. [39] developed a CNN for identifying the Time Delay of Arrival in micro-seismic events, which is essential for location determination and preventing mining disasters induced by a high-stress concentration. In addition to this micro-seismic event prediction, Derakhshani and Foruzan [40] developed DL neural networks to estimate seismic ground motion parameters. Another study on earthquake magnitude prediction has been carried out based on regression algorithms and cloud-based big data infrastructure [41]. In the mentioned study, it was shown that using big data analytics for predicting the magnitude of earthquakes opens a very promising research area and the methodology may enable simultaneous processing of massive data with a large number of variables.

### 3. Data Used

The training stage of an artificial intelligence algorithm is crucial and requires a sufficient amount of high-quality data. The datasets used for the training in this study have been downloaded from the freely available sources, image databases, and search engines listed in Table 1. A Python script has been implemented to download data from search engines (Google, Bing, Baidu) and Flickr, using specific keywords (i.e., *Landslides*, *Earthslip*, *Rockfall*, *Landslip*, *Rockslide*, and *Mudslide*) and their corresponding terms in 92 languages. A total of 36,984 images were downloaded from these sources and 600 of them were selected manually with visual checks by the expert to identify those which are most representative for different types of landslides. It has been observed that most landslide photos include roads, forest trails, mountains with vegetation cover, villages/settlement areas that are located in the foothills of mountains, landscapes, and other manmade objects. During the selection of the representative landslide photos, the completeness of the scene (variety of the object types obtaining the textural variation) was taken into account. For example, landslides that only showed mud were excluded and those which included mountains, forests, houses, etc., were kept. In principle, the landslide has to be one of the main themes or object types in the photo. In other words, photos where a landslide forms only a small part of the overall image were excluded. On the other hand, completely irrelevant photos (e.g., book cover pages, ads, selfies), post-disaster resilience photos without the actual landslide (e.g., with dozens of people, rescue teams, diggers), flooding photos, and images with a size smaller than 224 × 224 pixels were also disregarded while forming the training datasets.

**Table 1.** Photo sources for training the algorithm.

| Data source | Remarks |
| --- | --- |
| Google Images [42] | Search and download images with keywords: Landslides, Earthslip, Rockfall, Landslip, Rockslide, Mudslide (in 92 languages). |
| Flickr [43] | |
| Bing Images [44] | |
| Baidu Images [45] | |
| Places Dataset [46] | High-resolution images with a minimum dimension of 512 pixels. |
| Natural Image Database [47] | Nature Scene Collection and Human Made Scene Collection |

Six other object classes (*Irrelevant*, *Mountain-Valley*, *Natural*, *Road*, *Village*, and *Yard*) have been defined for the classification of the photos with the CNN algorithm. Apart from the irrelevant class, all others include natural scenes and have the potential to be mixed with the landslide during the classification. Examples of the training images of each class are given in Figure 2. In addition to the search engine and Flickr photos, Places [46] and Natural Image Database [47] datasets, which are distributed freely, were used for the training of the above-mentioned classes other than landslides. The Places Dataset was previously provided for the Places365-Challenge 2016 and includes high-resolution training, validation, and test images with a minimum dimension of 512 pixels. This data has been used for the training of the other six classes. From this Dataset, 5000 images for each class were selected initially, and a subset of them (600 for each class) were used in this study. The Natural Image dataset was employed for the estimation of missing image information (e.g., channels or occluded parts) and super-resolution images [48].

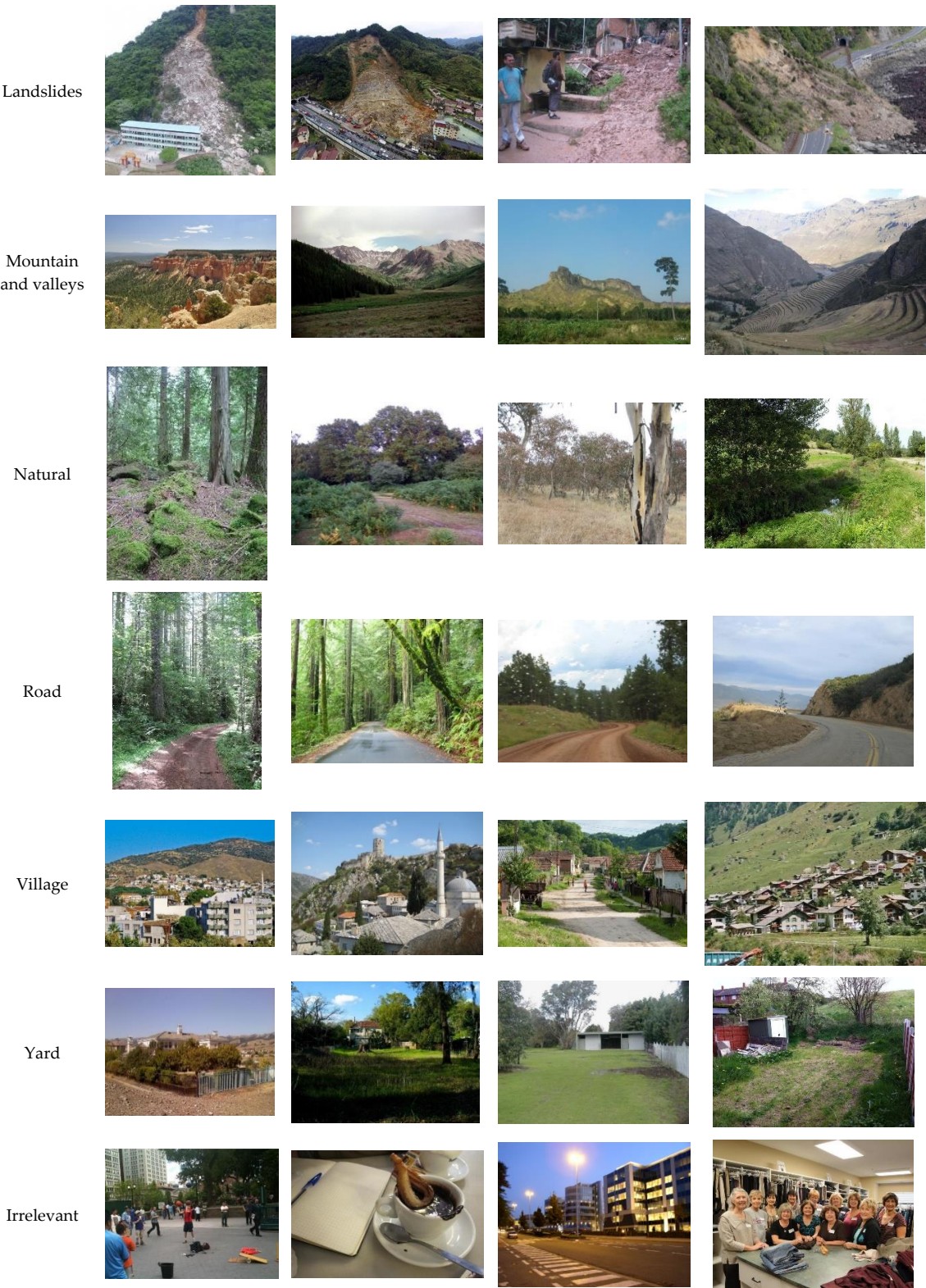

**Figure 2.** Examples of the photos used for training of each class type.

During the selection of the training photos in the other six classes, a number of principles were followed. First of all, the variation and semantic integrity in photos for each class type were ensured. The main focus/theme of the photo was identified (e.g., mountain and village) and photos with exceptional objects were excluded (i.e., generic road photos were preferred to photos of roads with

food stalls). The yard class photos were selected according to the combination of manmade objects with nature. The village and yard photos were selected with special care in order to distinguish them from the class *irrelevant*. Since the landslide dataset is composed of 500 training and 100 test photos (600 in total), datasets of other classes also include the same numbers of training and test photos.

Several problems were encountered during the data preparation. For example, several irrelevant photos for a given keyword were found. The number of useful landslides images among the downloaded ones was approximately 10% of the whole dataset. On the other hand, although the landslide photos were searched for in 92 languages, a significant amount of the downloaded images that were found were exactly the same. Additional processing was performed to remove the repetitive images from the same dataset. In addition, manual selection of the images by visual interpretations was one of the biggest challenges in this study.

## 4. Methodology

### 4.1. Data Preprocessing and Augmentation

An important part of the study was the derivation and preprocessing of training data, since no landslide class was available in the existing image databases. Therefore, each downloaded landslide image was assessed and classified manually. After selecting the images, two preprocessing steps were applied. First, the images were cropped and resized to $224 \times 224$ using the Python 3.7 programming language [49] and OpenCV computer vision library [50]. The cropping was applied to retain the aspect ratio of the pixels, as recommended by Zheng et al. [51], and thus increase the consistency of the classification. After reducing the image sizes, a data augmentation approach was applied to amplify the number of training images and thus to improve the suitability for the DL. The original training set includes 3500 images in total for all seven classes. Augmentation of the training set also helps to reduce the overfitting problem [52]. A Python Deep Learning library, Keras [53], was employed and modified to perform the CNN algorithm and tune it. The ImageDataGenerator class of Keras was used to augment the data and the best results were obtained by tuning the parameters to the following: (a) Rotation range: 15; (b) width & height shift ranges = 0.1; (c) horizontal flip: True; (d) fill mode: Nearest. In Figure 3, examples of two original landslide photos and their preprocessed and augmented versions are presented.

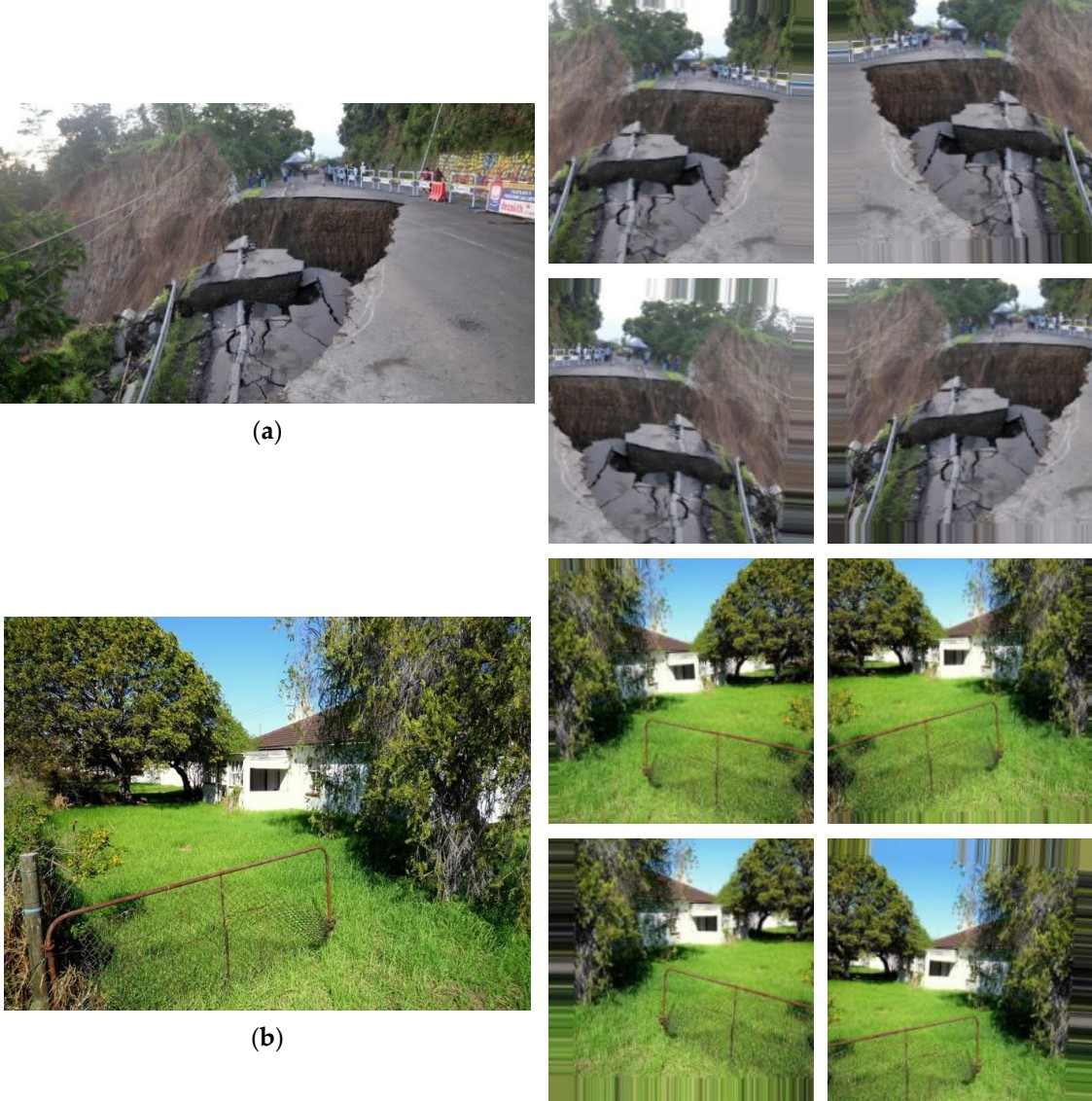

**Figure 3.** Two examples of the preprocessing and augmentation of the images. (**a**,**b**) are the original images, and the square photos are the cropped and augmented ones.

## 4.2. Photo Classification with the CNN

Girshick et al. [54] have proposed that the DL methods can be used through finetuning if the training dataset is small. Due to the small number of training images employed in this study, the VGG16 architecture proposed by Simonyan and Zisserman [55] was selected as the base model since it allows finetuning for CNN. The model was proposed by a team named VGG and placed in the top five in the localization and classification tracks of the ImageNet Challenge 2014 [56]. The VGG16 architecture was modified to increase the success rate of the classification of the landslide photos. The original VGG16 architecture and the version modified in this study are depicted in Figure 4. The modification mainly aims at finetuning of the model. The fully connected classifier part of VGG16 was removed from the model since it was trained using the ImageNet dataset [57]. The new fully connected classifier was initialized by using random values. All layers, apart from the fully connected classifier, were converted to non-trainable so that these initial random values could not update their weights during the training. In other terms, a fully connected classifier and thus, the random values, are not effective after back-propagation. This approach was defined as the first step of training and is shown in Figure 5. The r [58] (Equations (1)–(3)) and stochastic gradient descent (SGD) [59] (Equations (4) and

(5)) methods were selected as the optimizers in the first and second steps of the training, respectively. The SGD method has been found as the best and RMSprop as the best adaptive method by Wilson et al. [60] for CNN. The dense layer used in the architecture is a classic fully connected neural network layer, in which each input node is connected to each output node and the number of dense layers is seven because it equals the number of classes. Additionally, the parameter of the dense layer is 256 because it is connected to the previous layer and the previous layer's output is 256. Softmax used in the architecture is a loss function, which yields normalized class probabilities as outputs, while LeakyRelu is a variation of rectified linear units, which allows a non-zero gradient for the negative inputs [61]. One of the important components of the architecture is Dropout. Dropout, which was proposed by Srivastava et al. [62], is used for preventing overfitting. The data augmentation has been applied with a learning rate of 0.00001 and batch size of 50 using the resized dataset. The training has been stopped after 25 epochs (Figure 5) and the convolutional block 5 has been converted to trainable. This second step of training, as shown in Figure 6, activates back-propagation for the initial fully connected classifier and the fifth convolutional block.

$$v_t = \rho v_{t-1} + (1 - \rho) * g_t{}^2 \tag{1}$$

$$\Delta \omega_t = -\frac{\eta}{\sqrt{v_t + \varepsilon}} * g_t \tag{2}$$

$$\omega_{t+1} = \omega_t + \Delta \omega_t \tag{3}$$

where

$\eta$ : Initial Learning Rate

$v_t$ : Exponential Average of squares of gradients

$g_t$ : Gradient at time t along $\omega^j$

$$v_t = \beta v_{t-1} + (1 - \beta)\nabla_w L(W, X, y) \tag{4}$$

$$W = W - \alpha v_t \tag{5}$$

where

$\nabla_w$ : *Gradient with respect to weight and learning rate*

$\alpha$ : *Learning rate*

$\beta$ : *Momentum*

The model was compiled again in this step with a learning rate of 0.00001 and momentum of 0.9. Another 50 epochs of training were carried out using these parameters. After 25 epochs, the loss/accuracy ratio, generally, does not change. For this reason, 50 epochs are selected to terminate the learning stage. Hyperparameters were selected experimentally, considering the procedure recommended by Bengio [63]. In addition to the procedure recommended by Bengio [63], Masters and Luschi [64] showed that the combination of the small batch size and stochastic gradient descent make the training process stable and give good generalization performance. Consequently, the results of the experiments showed that the best performance was obtained when the batch size was 50, and the other hyperparameters were as shown in Figures 4 and 5.

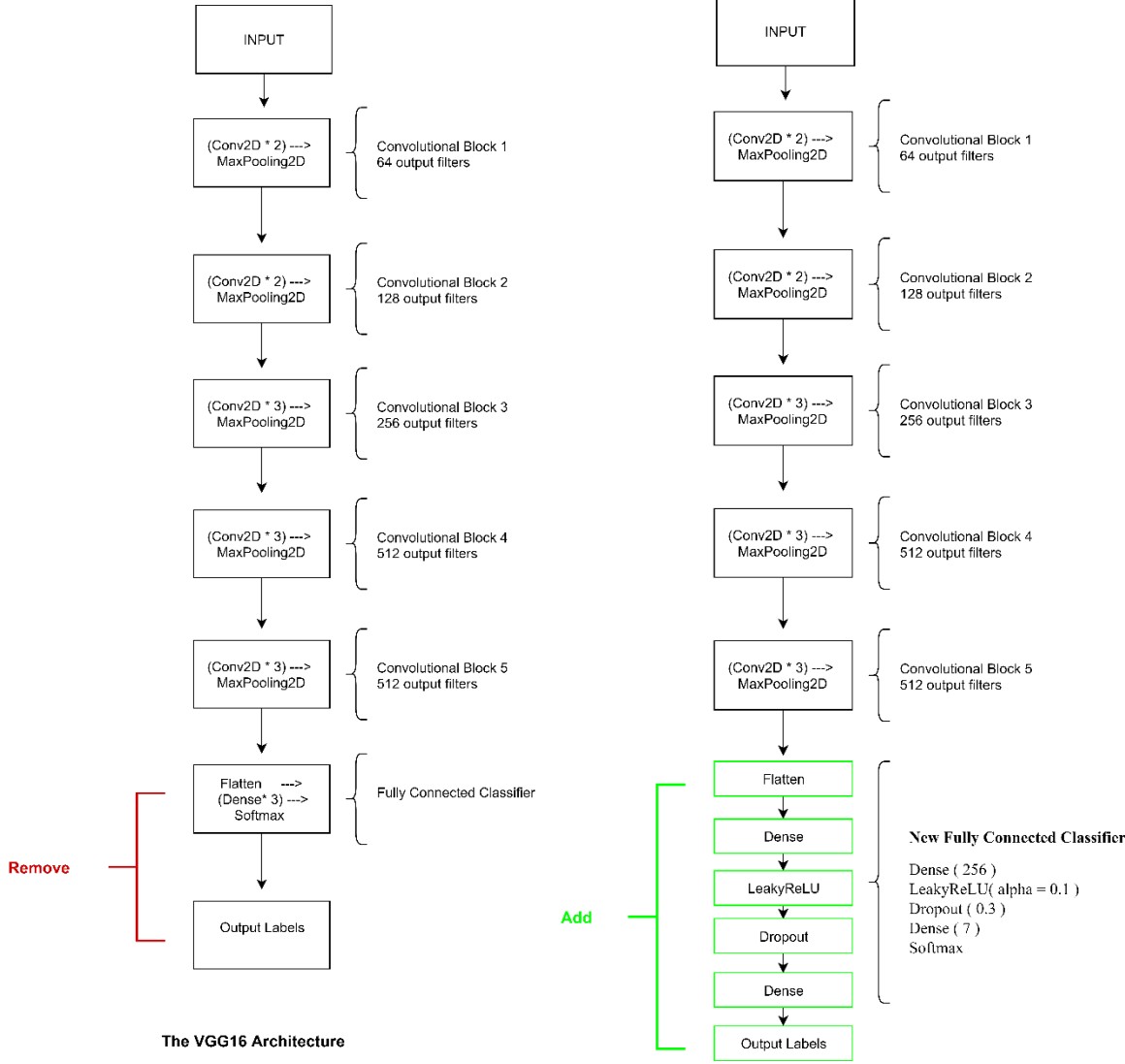

**Figure 4.** The original VGG16 [55] architecture (**left**) and the proposed architecture of this study (**right**).

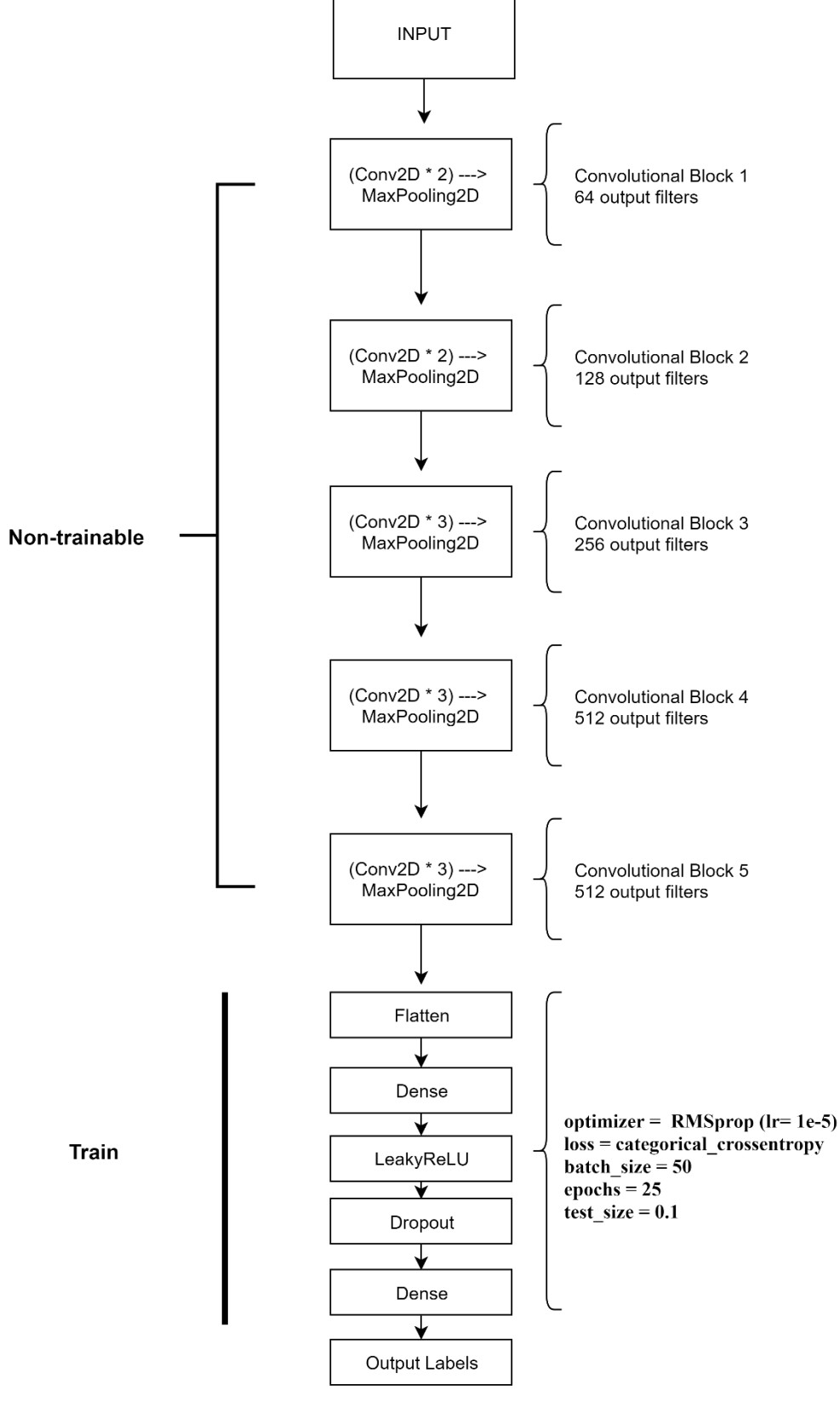

**Step 1**

**Figure 5.** The first step of training in the proposed algorithm.

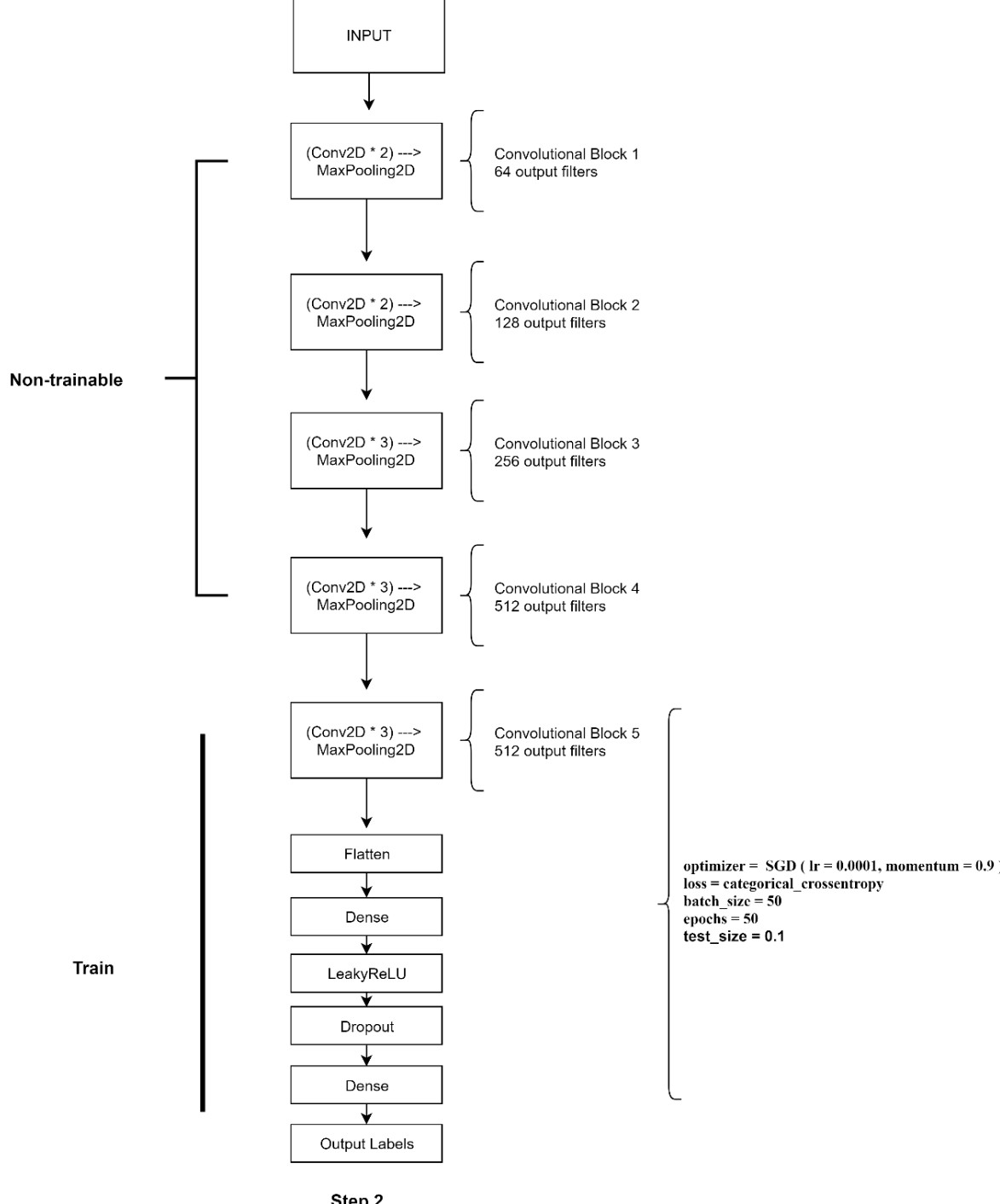

**Figure 6.** The second (final) step of the training in the proposed algorithm.

## 5. Results and Discussions

Ten different test runs were employed with ten sets of training and test datasets for the performance assessment. For each run, 10% of the images were used for the test and 90% were used for training. The training loss and accuracy results of each test are given in Figure 7. The model performances were assessed using the metrics function in Keras. The classification reports were generated using the classification report class in Python Scikit-learn library and the results of the forth run (Train-Test Split 4) are tabulated in Table 2.

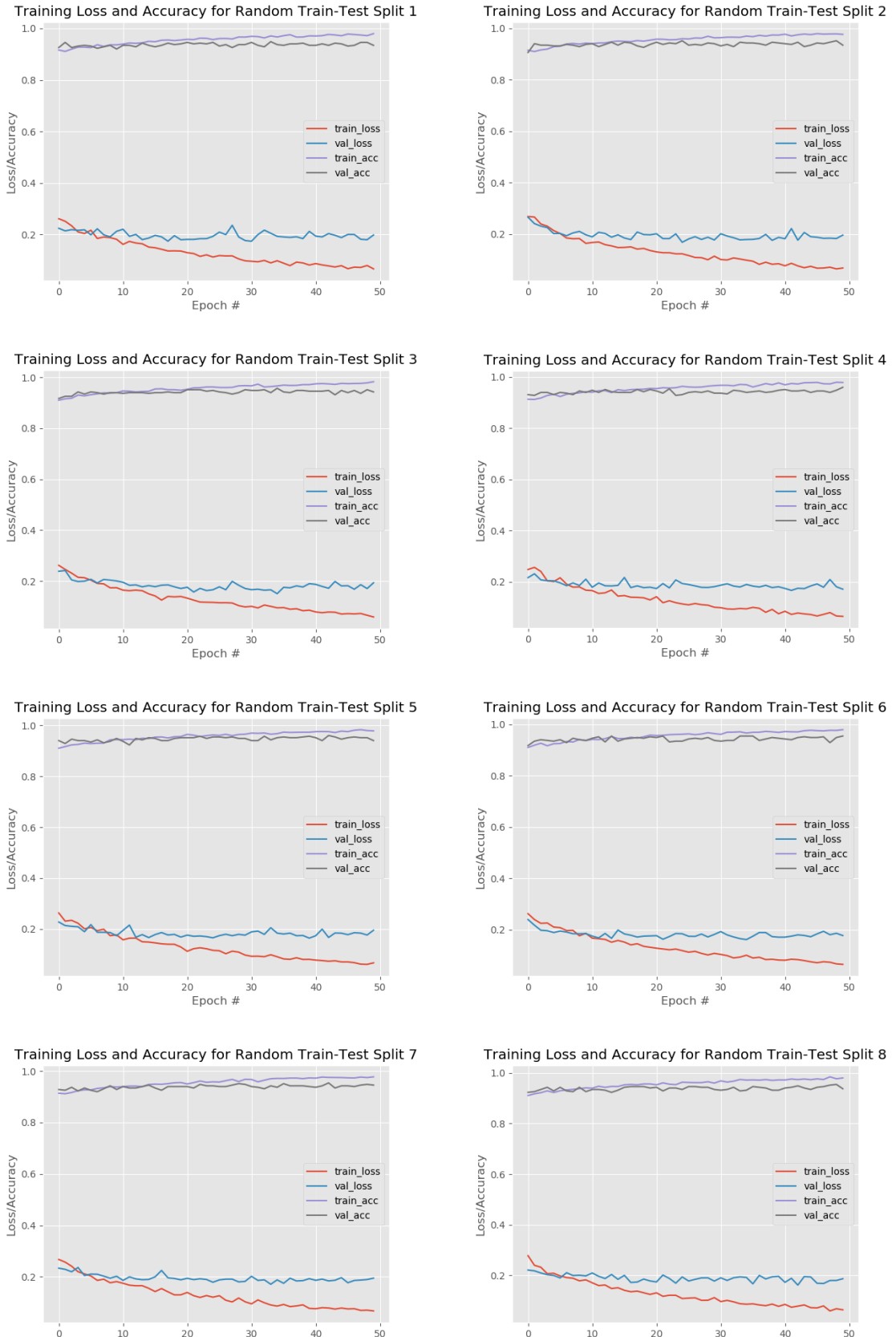

**Figure 7.** *Cont.*

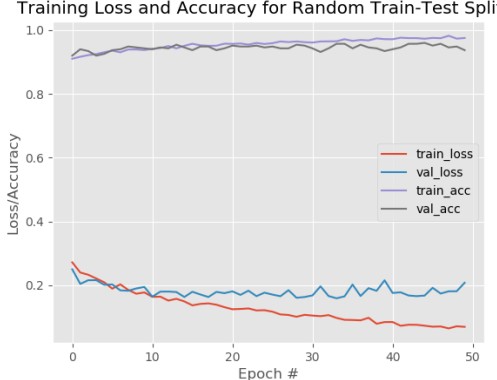
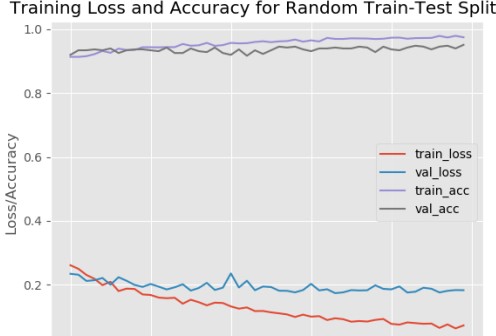

**Figure 7.** Training loss and accuracy results for the ten different runs of the model.

**Table 2.** The classification report of Train-Test Split 4 for all classes.

| Classes | Precision | Recall | F1-Score | Support |
|---|---|---|---|---|
| Irrelevant | 1.00 | 1.00 | 1.00 | 43 |
| Landslides | 0.94 | 0.95 | 0.94 | 62 |
| Mountain_valley | 0.98 | 0.98 | 0.98 | 49 |
| Natural | 0.95 | 0.91 | 0.93 | 45 |
| Road | 0.98 | 0.93 | 0.95 | 45 |
| Village | 0.93 | 0.98 | 0.96 | 44 |
| Yard | 0.95 | 0.97 | 0.96 | 62 |

Another test dataset with 700 images (100 for each class), which was not included in the training sets, was used for the evaluation of the Train-Test Split 4 model (forth test run). Additionally, an overfitting problem is not observed because the performance indicators of train and test datasets are close (Figure 7). The closest performance results are obtained from the Train–Test Split 4 model (forth test run), and hence, this model is selected. Use of Dropout in the CNN architecture helps to prevent the problem of overfitting. The normalized confusion matrix of this classification is presented in Figure 8. Some examples of correct and false classifications are given in the Appendix A. Table A1 in the Appendix A shows correct classifications for classes, while Table A2 in the Appendix A gives some false classifications.

As can be seen from the normalized confusion matrix (Figure 8), the CNN algorithm yields a high performance. This can be especially seen in the successful classification of irrelevant images by the algorithm. Additionally, landslide images were recognized with a high performance (96%). The lowest performance was obtained for natural and village images. In fact, this result can be expected, since it is extremely difficult to separate these classes. Despite this difficulty, the classification success is 85% for the images in this class. However, if the amount of training data was increased, the performance would be higher.

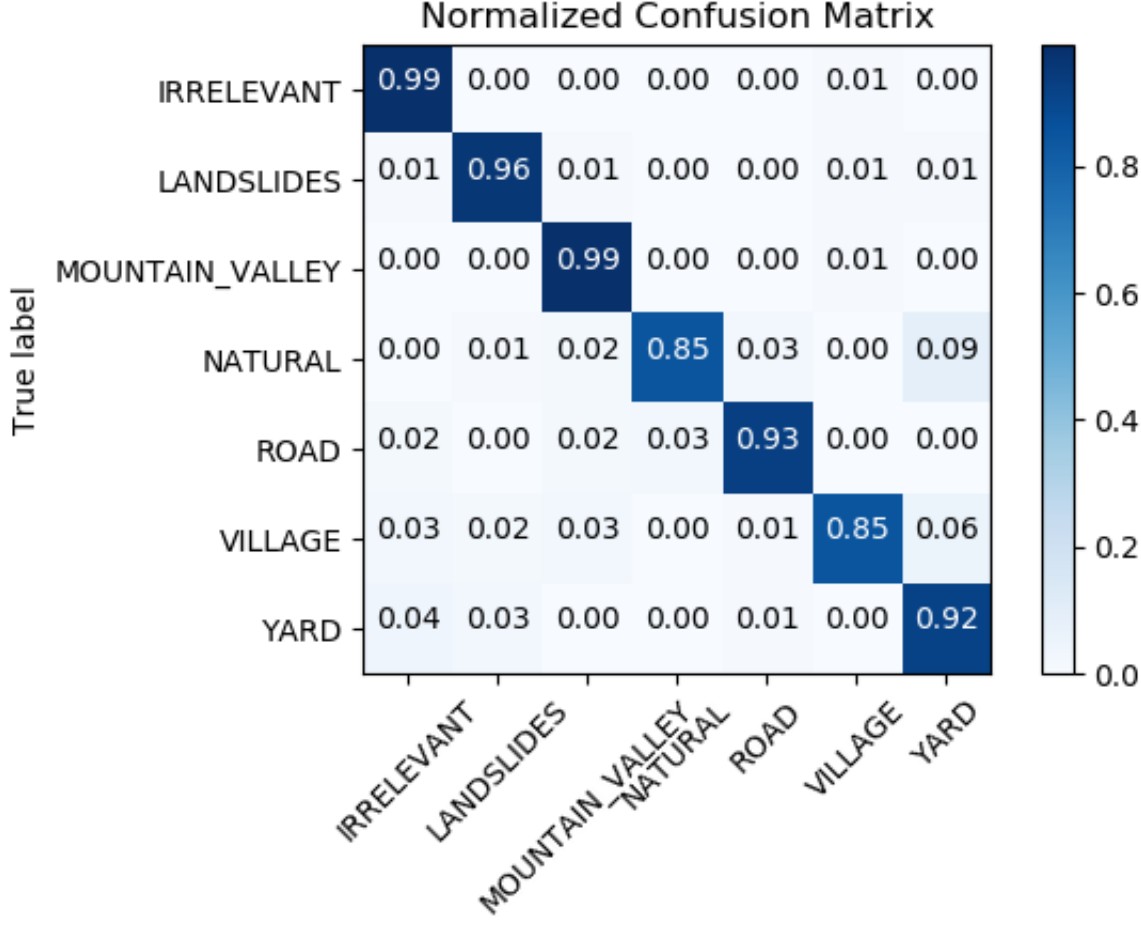

**Figure 8.** Normalized confusion matrix obtained from the forth test run (Train–Test Split 4 model).

## 6. Conclusions

Some serious paradigm changes in research methodologies have been observed during the last decade. CitSci is one of the results of these paradigm changes. In the near future, the volume of the data in almost every area of geoscience will increase drastically with the help of CitSci, since each ordinary person can act as a potential sensor. However, the quality of data provided by citizens or nonexperts should be checked automatically. With the advancements in artificial intelligence methods, the classification of photos from big datasets has become relatively easier. For specific classification purposes, suitable neural network architectures should be constructed. For the areas that utilize user-provided images, CNN provides great potential to identify the images. The amount of data is critical for the success of such architectures. An insufficient data volume results in overfitting, while a huge data volume affects the time efficiency and is a computational burden. Additionally, the complexity of the problem to be solved sometimes decreases the performance of the algorithm.

In the present study, a first attempt at classifying landslide photos using CNN was performed and promising results were obtained. The main difficulty encountered during the present study was data scarcity. To eliminate this problem, finetuning was applied to the CNN architecture, which is based on the VGG16 model. The architecture developed in the present study shows a high performance and the accuracy of the results is sufficient for the purpose. However, due to the small number of training images, the algorithm has a misclassification potential, and hence, the results should be checked manually until a sufficient amount of landslide photos is reached to train the algorithm further.

The proposed CNN architecture is part of a LS-CitSci study, where a mobile app called LaMA is employed for data collection together with a web map interface for uploading images and auxiliary

data for the use of volunteers and contributors who prefer not to download a mobile app. The web map interface also allows dynamic visualization of the uploaded data (www.geocitsci.com). The future work mainly involves: (i) Integration of the proposed quality check in the developed GIS platform; (ii) improving the performance by training the proposed algorithm using new landslide photos provided by the volunteers; and (iii) utilizing the collected CitSci data for natural hazard analysis and risk assessment purposes.

**Author Contributions:** Conceptualization, Sultan Kocaman and Candan Gokceoglu; Methodology, Recep Can, Sultan Kocaman and Candan Gokceoglu; Software, Recep Can; Validation, Recep Can and Candan Gokceoglu; Formal Analysis, Sultan Kocaman and Recep Can; Investigation, Recep Can and Sultan Kocaman; Resources, Sultan Kocaman and Candan Gokceoglu; Data Curation, Recep Can; Writing-Original Draft Preparation, Sultan Kocaman and Candan Gokceoglu; Writing-Review & Editing, Sultan Kocaman and Candan Gokceoglu; Visualization, Recep Can; Supervision, Sultan Kocaman and Candan Gokceoglu; Project Administration, Sultan Kocaman.

**Funding:** This research received no external funding.

**Conflicts of Interest:** The authors declare no conflict of interest.

## Appendix A

**Table A1.** Correct classifications by the CNN architecture developed in the study.

| Images | Actual & Predicted |
|---|---|
|  | Landslides |

**Table A1.** *Cont.*

| Images | Actual & Predicted |
|---|---|
| 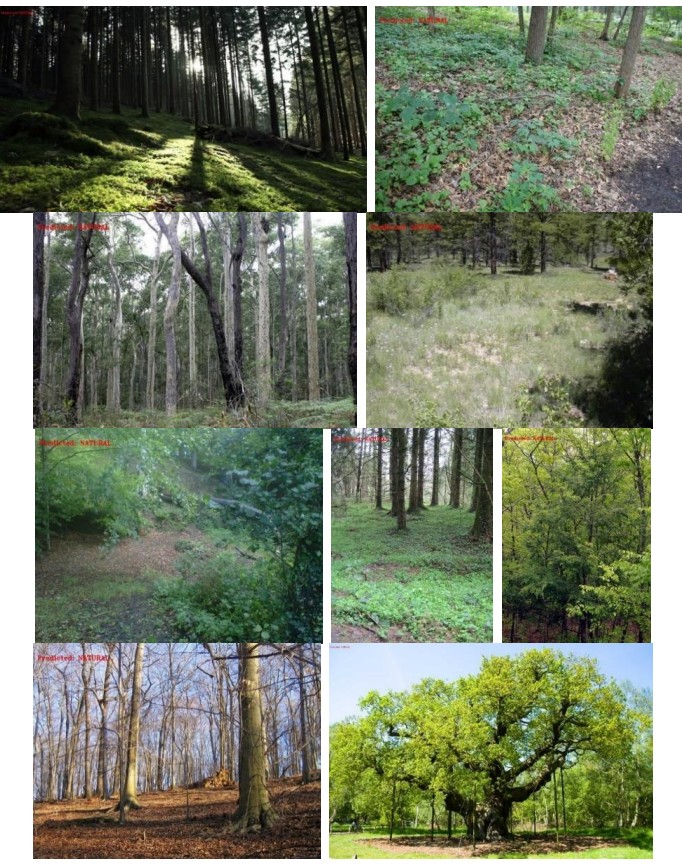 | Road |
|  | Natural |

**Table A1.** *Cont.*

| Images | Actual & Predicted |
|---|---|
|  | Yard |
|  | Irrelevant |

**Table A1.** *Cont.*

| Images | Actual & Predicted |
|--------|--------------------|
|  | Village |
|  | Mountain/ Valley |

**Table A2.** False classifications by the proposed CNN architecture.

| Images | Actual vs. Predicted |
| --- | --- |
| 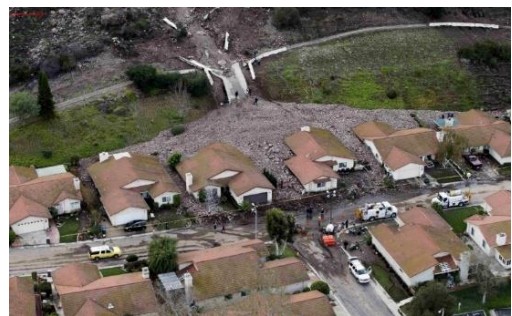 | Landslides vs. Village |
| 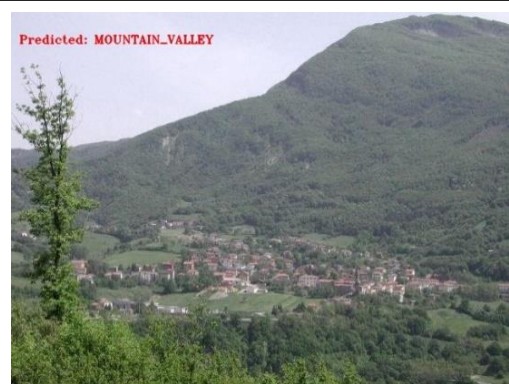 | Village vs. Mountain/Valley |
| 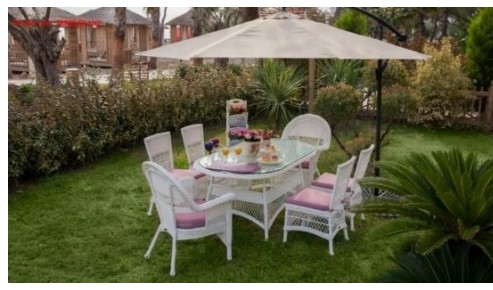 | Yard vs. Irrelevant |
| 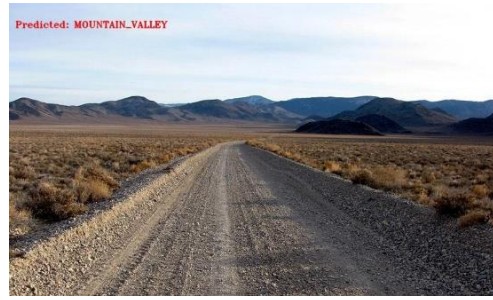 | Road vs. Mountain/Valley |

**Table A2.** *Cont.*

| Images | Actual vs. Predicted |
|---|---|
|  | Landslides vs. Irrelevant |
|  | Natural vs. Yard |
|  | Natural vs. Yard |
|  | Natural vs. Road |
|  | Village vs. Landslides |

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
