# Peer review of "A Convolutional Neural Network Architecture for Auto-Detection of Landslide Photographs to Assess Citizen Science and Volunteered Geographic Information Data Quality"

_ijgi, doi:10.3390/ijgi8070300_

Round 1

Reviewer 1 Report

## Specific comments
**1) Introduction**
What about the natural hazard and risk management question ?

*l. 13-14*: Citizen  Science goes back to the 90's, it's not a new term and not related to (geospatial) technologies development.
*l. 40*: CS and VGI are definitely not the same. (cite: Haklay, 2013?)
*l. 44-53*: the paragraph is too much self-centered, more general literature about landslides should be investigated and how it's incorporated or linked to natural hazard and risk management.
*l. 69-73*: a nice beginning of the methodology, but unfortunately it stopped here.

**2) Data employed**
*l. 76*: which datasets ? Does an open repository exist for the dataset build in this study (address the question of reproducibility) ?
*l. 80*: 92 languages; nice internationalization, but how were the translations made?
*l. 81*: why only 600 images ?
*l. 81-21*: what are the most representative images? And why are they used? This can induce some overfitting in the results of the ML algorithm.
*l. 81*: what is the visual analysis? It's not described.
*l. 97*: why these six others classes?
*Fig. 1*: 'mountain and valleys' are one single class or two separate? 'Nature' and 'yard' classes are probably too broad and with too much overlap. Do these two classes really need to be separated?
*l. 105-106*: based on which criterion were the 5'000 and 600 subset images selected? What was the subset used for?
*l. 111*: how the variation and semantic integrity in pictures has been ensured?
*l. 115*: what special care has been taken in selecting the images and why?
*l. 123-124*: How? By citizen scientists? Experts? A panel of both?

**3) Methodology**

**3.1 CNN studies**
This section is technically a review of SotA, not the description of the methodology and it's not well structured; there is few to no consistency between the paragraphs.
In addition, it doesn't place the current paper in the current research landscape.

**3.2 Data pre-processing and augmentation**
*l. 204*: classified manually: how?
*l. 214*: how did were these "best parameters" found?

**3.3 Photo classification with the CNN**
Generally too few details on how hyperparameters were set.
What innovation does the proposed architecture bring?
How were both the learning rates set? And why it's the same for the two steps?

*l. 223*: why was the VGG architecture used and not some other architectures? Were other architectures explored as well? VGG is dating from 2014, isn't there more robust and modern ones?
*l. 230*: how were the random values set?
*l. 235*: why two different optimizers here?
*l. 238-242*: can't it be resumed in a table?
*l. 237 and 241*: feel free to use scientific notation on learning rates.

**4) Application and Performance Assessment**
*l. 254*: why ten tests?
*l. 255*: why a 10-90 split? Is that better than 80-20 or 70-30?
*l. 259*: why were the 4th run specially selected?
*Fig. 6*: there is obviously overfitting in every steps. Caption and legends are not consistent and not clear.
*Table 2*; no explanation of the last 3 lines (and all the numbers are the same. Why? What are these numbers?).
*l. 282*: what is the other dataset? Where does it come from?
*l. 291*: as it is extremely difficult to separate these classes, why were they chosen?

In general, there is too few images for training a ML model.
The question: "How to be sure the learned features are relevant?" is not addressed.

In addition, computation environment and computation times should be reported for the training stages as well as for the inference stage.

**5) Conclusion**
The application seem promising but were not found in the play store.

*l. 302*: this is somehow what this study has proven.
*l. 305*: how to solve for data scarcity? Probably using citizen science where citizen are involved in the project.
*l. 311-313*: that is what was expected from this study (according to its title) but never found in the methodology.
---
# Original article structure:

## A Convolutional Neural Network Architecture for Auto-Detection of Landslide Photographs to Assess Citizen Science and VGI Data Quality

### Abstract

### Introduction

### Data Employed

### Methodology
#### CNN Studies

#### Data Pre-processing and Augmentation

#### Photo classification with the CNN

### Application and Performance Assessment

### Conclusion

Author Response

Dear Reviewer 1,

Thank you very much for your valuable comments. We believe that the quality of the manuscript has been improved significantly after the revision. We hope that the revised version of the manuscript may satisfy the Reviewer.

Please find our answers to your comments below.

Kind regards,

Authors

## Specific comments
**1) Introduction**
What about the natural hazard and risk management question ?

Authors: The main purpose of the study is to check the landslide photos uploaded by volunteers and non-experts. However, CitSci has a great potential to contribute natural hazard and risk management efforts. These were described in the previous studies cited in the manuscript [10, 15, 17 and 18]. For this reason, these are not given again in the manuscript. We hope that this explanation is sufficient for the Reviewer.

*l. 13-14*: Citizen  Science goes back to the 90's, it's not a new term and not related to (geospatial) technologies development.

Authors: We have revised the sentence accordingly. We thank to the Reviewer for his/her valuable comment.

*l. 40*: CS and VGI are definitely not the same. (cite: Haklay, 2013?)

Authors: Reference is added and necessary correction is penned.

*l. 44-53*: the paragraph is too much self-centered, more general literature about landslides should be investigated and how it's incorporated or linked to natural hazard and risk management.

Authors: In fact, the present study focuses on checking photos uploaded by citizens. However, we should mention our previous studies because these recent studies are also closely related to this study. Our previous studies define the potential benefits and use of CitSci in natural hazard and risk management, especially landslide. In addition, this study provides a solution for the problem mentioned by Kocaman and Gokceoglu [18]. The use of CitSci in landslide researches is a new topic and it is impossible to find more studies in international literature except NASA’s Landslide Reporter. We hope that this explanation may satisfy the Reviewer.

*l. 69-73*: a nice beginning of the methodology, but unfortunately it stopped here.

Authors: We thank to the Reviewer for this valuable comment and careful assessment. We revised this part according to his/her comment. We think that the revised version of the manuscript is more understandable.

**2) Data employed**
*l. 76*: which datasets ? Does an open repository exist for the dataset build in this study (address the question of reproducibility) ?

Authors: The repository was compiled using various search engines and social media platform images as described in the manuscript and briefly listed in Table 1 by the authors. It could be provided freely in the future upon request.

*l. 80*: 92 languages; nice internationalization, but how were the translations made?

Authors: Translations were carried out via embedding an online translation tool into the developed code. (https://translatr.varunmalhotra.xyz/)

*l. 81*: why only 600 images ?

Authors: The number is relatively small since the most representative images were used for the training of each class.

*l. 81-21*: what are the most representative images? And why are they used? This can induce some overfitting in the results of the ML algorithm.

Authors: We would like to explain this with an example. The following photo (a) is considered as one of the most representative photos for a landslide event. The same scene could be imaged as in (b), (c) or (d). However, the latter ones could also be mistaken with other activities, such as a construction site, group of people having sight seeing, etc. On the other hand, the photo in (a) definitely describes a landslide event scene without further doubt or discussion. Therefore, photos such that a human mind would confidently say that “a landslide has occured here” are selected as the most representative ones.

 (a)

 (b)

 (c)

 (d)

*l. 81*: what is the visual analysis? It's not described.

Authors:The phrase is revised as “visual checks of the expert”.

*l. 97*: why these six others classes?

Authors:The classes are defined empirically and instead of binary classification of the images as landslide and non-landslide (irrelevant), this class scheme has been kept to understand the correlations between classes and also find out other classification possibilities.

*Fig. 1*: 'mountain and valleys' are one single class or two separate? 'Nature' and 'yard' classes are probably too broad and with too much overlap. Do these two classes really need to be separated?

Authors: “Mountain and valleys” is the name of one single class, and this title has been given since the majority of the training images in this class involve both earth forms.Nature and yard images have been classified seperately to see the possibility of classification, and as can be seen from Figure 7, they have the highest value in the confusion matrix (0.09). Still, it has been possible to classify them separately with a high performance.

*l. 105-106*: based on which criterion were the 5'000 and 600 subset images selected? What was the subset used for?

Authors: 5000 images have been selected manually with visual interpretation, and the most representative ones (600 of them) have been used for the training.

*l. 111*: how the variation and semantic integrity in pictures has been ensured?

Authors: We have trusted the experience of the authors for this purpose.

*l. 115*: what special care has been taken in selecting the images and why?

Authors: We have revised the sentence.

*l. 123-124*: How? By citizen scientists? Experts? A panel of both?

Authors: The authors have carried out the visual interpretation and manual selection process.

**3) Methodology**

**3.1 CNN studies**
This section is technically a review of SotA, not the description of the methodology and it's not well structured; there is few to no consistency between the paragraphs.
In addition, it doesn't place the current paper in the current research landscape.

Authors: We have revised the main structure of the manuscript, and renamed and moved this part under a separate section after the Introduction (2. Previous Studies on the Use of CNN in Geosciences). We think that the readibility of the manuscript has been improved after the re-structuring. In addition, the Methodology Section (4) has been enhanced significantly.

**3.2 Data pre-processing and augmentation**
*l. 204*: classified manually: how?

Authors: Although the images were downloaded using the keyword “landslides”, the downloaded images included irrelevant images like selfies, book covers, etc. Therefore, the authors manually classified and labelled the relevant images by visually checking and interpreting each downloaded image.

*l. 214*: how did were these "best parameters" found?

Authors: The best parameters have been chosen by manual tuning of the parameters experimentally through assessing their performances after each run. The initial values for the parameters have been assigned randomly by using the initializer function provided by Keras. The manuscript has also been revised to include this explanation.

**3.3 Photo classification with the CNN**
Generally too few details on how hyperparameters were set.
What innovation does the proposed architecture bring?

Authors: Hyperparameters were set based on the iterative process done manually by the authors.The process includes choosing the parameter and observation of each epoch.There is no architecture available in the literature that auto detects the landslide photographs, so the proposed architecture forms the novelty of the study.

How were both the learning rates set? And why it's the same for the two steps?

Authors: It is actually different for both steps.The parameters were chosen by evaluating the performance of the model for each selected value.The changes in the values have been observed to see the value ranges that ensure better performance.

*l. 223*: why was the VGG architecture used and not some other architectures? Were other architectures explored as well? VGG is dating from 2014, isn't there more robust and modern ones?

Authors: Since the VGG architecture achieved the best score on scene classification task, the authors decided to choose VGG architecture (for further explanation, please refer to http://places2.csail.mit.edu/PAMI_places.pdf / 5.5.). Another reason for not using any other architecture is that the proposed architecture is the first one in the literature and yet quite successful. In addition, the dataset that were collected and labelled by the authors are also the first and there is no dataset available in the literature. However, there may be room for further research and future studies with other architectures may obtain better results.

*l. 230*: how were the random values set?

Authors: Using initializer class provided by Keras library. (https://keras.io/initializers/)

*l. 235*: why two different optimizers here?

Authors: It is answered in 243-246.

*l. 238-242*: can't it be resumed in a table?

Authors: We think that a table might be more confusing.

*l. 237 and 241*: feel free to use scientific notation on learning rates.

Authors: 1e-5

**4) Application and Performance Assessment**
*l. 254*: why ten tests?

Authors: Since the dataset were splitted as %10 test and %90 train, ten different set ensured the use of every image for both training and test image at different runs. If the dataset variation is good and balanced, all the models should give same accuracy-loss graphics.

*l. 255*: why a 10-90 split? Is that better than 80-20 or 70-30?

Authors: Because of the dataset size.

*l. 259*: why were the 4th run specially selected?

Authors: This model provides the minimum difference between training and test. Therefore this model has been selected for further accuracy investigations.

*Fig. 6*: there is obviously overfitting in every steps. Caption and legends are not consistent and not clear.

Authors: We don’t agree with the Reviewer. The train-test accuracy graphs given in Figure 7 (old Figure 6) clearly shows that no overfitting has occured.

*Table 2*; no explanation of the last 3 lines (and all the numbers are the same. Why? What are these numbers?).

Authors: We have removed the lines from the Table.

*l. 282*: what is the other dataset? Where does it come from?

Authors: The test dataset has been created from the downloaded images as explained above in order to evaluate how the model performs for unseen data.

*l. 291*: as it is extremely difficult to separate these classes, why were they chosen?

Authors: Because if it is searched for where the landslides occurs, it is easily seen that both village and nature are subject to landslides and have a potential that volunteers can send images from there.

In general, there is too few images for training a ML model.
The question: "How to be sure the learned features are relevant?" is not addressed.

Authors: There is one source that the model can be tested and evaluated for ensuring the model is doing right apart from our dataset is citizens.If they use the platform effectively and we can collect more data, we can be sure that the learned features are relevant.

In addition, computation environment and computation times should be reported for the training stages as well as for the inference stage.

Authors: Since we have a small training set, computation time is very short and therefore irrelevant.

**5) Conclusion**
The application seem promising but were not found in the play store.

Authors: The app can be found on the playstore when searched with “Landslide monitoring app”. The first row in the query result is our app named LaMA.

*l. 302*: this is somehow what this study has proven.

Authors: Please see our previous answers.

*l. 305*: how to solve for data scarcity? Probably using citizen science where citizen are involved in the project.

Authors: That is part the future work as mentioned in the Conclusion in the revised version of the manuscript.

*l. 311-313*: that is what was expected from this study (according to its title) but never found in the methodology.

Authors: That is the main functionality of the developed approach in the whole LS-CitSci project, i.e. automatic assessment of CitSci and VGI data quality. Therefore, we see a need to keep this part in the title.

Reviewer 2 Report

Dear authors, 

I have seen the manuscript and seen that this paper dealt with important topic of landslides which is one of the serious natural hazards. It is an interesting paper, I would suggest to publish this paper with some major comments as below:

Abstract should be modified and improved, in my opinion, the line 12 - 19, page 1 should be removed in the Abstract, it might be information of Introduction part, not in the Abstract. Then, please modify the Abstract accordingly. 

Introduction is simple and not interesting, the authors are requested to include more recent works on landslides using advanced techniques like Machine learning methods (Refer and cite papers of Dr Binh Thai Pham https://www.researchgate.net/profile/Binh_Pham10)

Objective of the paper is not clear stated in the Introduction, the authors are requested to provide the novelty of this research, what are new and different works compared with published papers.

Figures are not clear enough for publication in the current format, please improve them.

I did not see the Results and Discussion parts, please separate these two sections and modify them accordingly. 

I did not see enough discussion in this paper, please discuss on the finding of the works and compared them with other published works.

Conclusion parts is qualified enough, please conclude on the finding of the works only, and remove the unnecessary information, please give the limitation and future works of this paper as well

Paper is readable, but please double check the English of the paper by Native English experts. 

Good luck!

Author Response

Dear Reviewer,

Thank you very much for your valuable comments. We believe that the quality of the manuscript has been improved significantly after the revision. We hope that the revised version of the manuscript may satisfy the Reviewer.

Please find our answers to your comments below.

Kind regards,

Authors

Dear authors, 

I have seen the manuscript and seen that this paper dealt with important topic of landslides which is one of the serious natural hazards. It is an interesting paper, I would suggest to publish this paper with some major comments as below:

Abstract should be modified and improved, in my opinion, the line 12 - 19, page 1 should be removed in the Abstract, it might be information of Introduction part, not in the Abstract. Then, please modify the Abstract accordingly. 

Authors: Thank you very much for the recommendation. We have revised the abstract substantially.

Introduction is simple and not interesting, the authors are requested to include more recent works on landslides using advanced techniques like Machine learning methods (Refer and cite papers of Dr Binh Thai Pham https://www.researchgate.net/profile/Binh_Pham10)

Authors: Thank you very much for suggestion. We have added few references on the use ML methods for landslide susceptibility mapping to the manuscript.

Objective of the paper is not clear stated in the Introduction, the authors are requested to provide the novelty of this research, what are new and different works compared with published papers.

Authors: Thank you very much for your valuable input. We have highlighted the novelty of the research in the last two paragraphs of the Introduction. Since the use of citizen science and VGI for landslide researches is a new research field, there are no studies to compare.

Figures are not clear enough for publication in the current format, please improve them.

Authors: Thank you. We have improved the figures.

I did not see the Results and Discussion parts, please separate these two sections and modify them accordingly. I did not see enough discussion in this paper, please discuss on the finding of the works and compared them with other published works.

Authors: Thank you. We have modified the structure of the manuscript and added the Results and Discussions section (5). We have improved the discussions as well.

Conclusion parts is qualified enough, please conclude on the finding of the works only, and remove the unnecessary information, please give the limitation and future works of this paper as well

Authors: Thank you for the recommendation. We have modified the Conclusions and added a sentence for future work.

Paper is readable, but please double check the English of the paper by Native English experts. 

Good luck!

Reviewer 3 Report

I have read and carefully evaluated this manuscript that addressed an interesting topic. However, the manuscript needs some further improved before to be accepted for publication. My suggestions are as follows:

- In the Abstract, the methodology used should be indicated. Also, clearly mention the benefits of your study and findings.

- There are occasional grammar errors through the manuscript especially the article ''the'', ''a'' or ''an'' is missing or redundant in many places, please make a spellchecking in addition to these minor issues. For example:

Line 29: drop "the" before citizen science.

Line 60: add "the" before veracity.

Line 65: add "the" before increase and before amount.

Line 74: Data Used.

Line 105: add "the" before other.

Line 137: add "the" after due to.

Line 146: add "a" before massive.

Line 152: add "a" before low.

Line 169: replace "with training of" by "using".

Line 170 add drop "," after although. 

Line 172 add "a" before landslide.

Line 223-224: Bad English! Please revise.

Line 152: delete "s" of training.

Line 256: Replace "is" with "are".

Line 285-286: Add "the" before Appendix.

- Figures 3-5 are of low quality.

- Line 256: What do you mean by "training loss"?

- How did you "add a fine tuning to the CNN"? You just tuned the architecture by selecting the best hyper-parameters. Please revise the sentence.

Author Response

Dear Reviewer,

Thank you very much for your valuable comments. We believe that the quality of the manuscript has been improved significantly after the revision. We hope that the revised version of the manuscript may satisfy the Reviewer.

Please find our answers to your comments below.

Kind regards,

Authors

I have read and carefully evaluated this manuscript that addressed an interesting topic. However, the manuscript needs some further improved before to be accepted for publication. My suggestions are as follows:

- In the Abstract, the methodology used should be indicated. Also, clearly mention the benefits of your study and findings.

Authors: Thank you very much for the recommendation. We have revised the abstract substantially.

- There are occasional grammar errors through the manuscript especially the article ''the'', ''a'' or ''an'' is missing or redundant in many places, please make a spellchecking in addition to these minor issues. For example:

Authors: Thank you. We have checked the manuscript carefully and revised the use of articles. In addition, the following recommendations have also been applied to the manuscript.

Line 29: drop "the" before citizen science.

Line 60: add "the" before veracity.

Line 65: add "the" before increase and before amount.

Line 74: Data Used.

Line 105: add "the" before other.

Line 137: add "the" after due to.

Line 146: add "a" before massive.

Line 152: add "a" before low.

Line 169: replace "with training of" by "using".

Line 170 add drop "," after although. 

Line 172 add "a" before landslide.

Line 223-224: Bad English! Please revise.

Authors: Thank you. We have revised the sentence as ‘Due to the small number of training images employed in this study, the VGG16 architecture proposed by Simonyan and Zisserman [47] has been selected as base model since it allows fine-tuning for CNN.’

Line 152: delete "s" of training.

Authors: Thank you. We have revised as proposed.

Line 256: Replace "is" with "are".

Authors: Thank you. We have revised as proposed.

Line 285-286: Add "the" before Appendix.

Authors: Thank you. We have revised as proposed.

- Figures 3-5 are of low quality.

Authors: Thank you. We have improved the quality of the figures.

- Line 256: What do you mean by "training loss"?

Authors: Thank you. The training loss means the number indicating how bad the models prediction was on training.

- How did you "add a fine tuning to the CNN"? You just tuned the architecture by selecting the best hyper-parameters. Please revise the sentence.

Authors: Thank you. We have revised the sentence.

Round 2

Reviewer 1 Report

I'm still thinking that the paper shows overfitting - I would suggest that the authors check the following article : Srivastava, N., Hinton, G., Krizhevsky, A., Sutskever, I., and Salakhutdinov, R. (2014).Dropout: A simple way to prevent neural networks from overfitting.Journal of MachineLearning Research,15, 1929–195

Author Response

Dear Reviewer,

Thank you for your suggestion.

We have already considered Dropout algorithm in our architecture to prevent from overfitting problem. Please see Lines 283-284 in the manuscript (One of the important components of the architecture is Dropout. Dropout, which was proposed by Srivastava et al [62], is used for preventing overfitting.).

Figures 4, 5 and 6 also show that the Dropout algorithm from the mentioned article (Srivastava, N., Hinton, G., Krizhevsky, A., Sutskever, I., and Salakhutdinov, R. (2014).Dropout: A simple way to prevent neural networks from overfitting.Journal of MachineLearning Research,15, 1929–195) has already been employed in our architecture.

Additionally, we checked Srivastava et al. (2014) paper again. We concluded that the accuracy of our CNN model is sufficient for our purpose. We think that the model will be much more accurate depending on the increase of the training data. In Figure 7, the training and validation losses; and the training and validation accuracies are quite similar. The vertical axis is in the decimal level as well.

Kind regards,

Reviewer 2 Report

It is my pleasure to accept your paper. 

Thanks! 

Author Response

Dear Reviewer,

Thank you for your valuable contributions to our manuscript and considering it for publication.

Kind regards

Authors